# Effect of topical emollient oil application on weight of preterm newborns: A systematic review and meta-analysis

Fekadeselassie Belege Getaneh[1]*, Anissa Mohammed[1], Alemu Gedefie Belete[1], Amare Muche[1], Aznamariyam Ayres[1], Yibeltal Asmamaw[1], Zemen Mengesha[1], Asrat Dimtse[2], Natnael Moges Misganaw[3], Dires Birhanu Mihretie[4], Zebenay Workneh Bitew[5], Meaza Mengstu[1], Asressie Molla[1]

1 College of Medicine and Health Sciences, Wollo University, Dessie, Ethiopia, 2 College of Health Sciences, Addis Ababa University, Addis Ababa, Ethiopia, 3 College of Health Sciences, Debre Tabor University, Debre Tabor, Ethiopia, 4 College of Health Sciences, Dilla University, Dilla, Ethiopia, 5 College of Health Sciences, St. Paul's Hospital Millennium Medical College, Addis Ababa, Ethiopia

* fekadebelege@gmail.com, fekadeselassie.belege@wu.edu.et

## Abstract

### Background

Synthesizing current evidence on interventions to improve survival outcomes in preterm infants is crucial for informing programs and policies. The objective of this study is to investigate the impact of topical emollient oil application on the weight of preterm infants.

### Methods

A systematic review and meta-analysis of randomized controlled trials (RCTs) was conducted. To identify relevant studies, comprehensive searches were conducted across multiple databases, including PubMed, Cochrane, Scopus, Clinical trials, ProQuest Central, Epistemonikos, and gray literature sources. The inclusion criteria were based on the PICO (Population, Intervention, Comparison, and Outcomes) format. Study quality was assessed using the Cochrane risk of bias tool for randomized trials (RoB 2.0). Data analysis was performed using StataCrop MP V.17 software, which included evaluating heterogeneity, conducting subgroup analysis, sensitivity analysis, and meta-regression. The findings were reported in accordance with the PRISMA checklist, and the review was registered with PROSPERO (CRD42023413770).

### Results

Out of the initial pool of 2734 articles, a total of 18 studies involving 1454 preterm neonates were included in the final analysis. Fourteen of these studies provided data that contributed to the calculation of the pooled difference in mean weight gain in preterm neonates. The random effects meta-analysis revealed a significant pooled difference in mean weight gain of 52.15 grams (95% CI: 45.96, 58.35), albeit with high heterogeneity ($I^2 > 93.24\%$, p 0.000). Subgroup analyses were conducted, revealing that preterm infants who received massages

**Funding:** The author(s) received no specific funding for this work.

**Competing interests:** The authors have declared that no competing interests exist.

**Abbreviations:** Aes, Assess adverse events; AHRQ, Agency for Healthcare Research and Quality; CI, Confidence intervals; DECIDE, Developing and Evaluating Communication strategies to support Informed Decisions and practice based on Evidence; LBW, Low birth weight; MCT, Medium chain triglycerides; MD, Mean difference; NICU, Neonatal intensive care unit; PRISMA, Preferred Reporting Items for Systematic Reviews and Meta-Analysis; RCTs, Randomized control trials; RR, Risk ratio; UNICEF, United Nations International Children's Emergency Fund; WHO, World Health Organization.

three times daily with either sunflower oil or coconut oil exhibited greater mean differences in weight gain. Meta-regression analysis indicated that the type of emollient oil, duration of therapy, and frequency of application significantly contributed to the observed heterogeneity. A sensitivity analysis was performed, excluding two outlier studies, resulting in a pooled mean weight difference of 78.57grams (95% CI: 52.46, 104.68). Among the nine studies that reported adverse events, only two mentioned occurrences of rash and accidental slippage in the intervention groups.

## Conclusion

The available evidence suggests that the application of topical emollient oil in preterm neonates is likely to be effective in promoting weight gain, with a moderate-to-high level of certainty. Based on these findings, it is recommended that local policymakers and health planners prioritize the routine use of emollient oils in newborn care for preterm infants. By incorporating emollient oils into standard care protocols, healthcare providers can provide additional support to promote optimal growth and development in preterm infants.

## Introduction

### Background

Preterm births refer to neonatal births occurring before the completion of 37 weeks of gestation [1]. It is estimated that Asia and sub-Saharan Africa account for 81.1% of preterm births and over 80% of all newborn mortality among preterm neonates [2]. One of the challenges faced by preterm newborns is their underdeveloped skin, which lacks an adequate epidermal barrier. The stratum corneum, responsible for regulating the function of the epidermal barrier, typically reaches functional maturity in normal development around 32 to 34 weeks of gestation [3]. However, in preterm infants, this skin barrier is often compromised, leading to high rates of trans-epidermal water loss and simultaneous losses of fluid and heat [4–6].

In 2014, the World Health Organization (WHO) and UNICEF introduced Every Newborn Action Plan (ENAP) as a global roadmap to reduce avoidable newborn mortality and stillbirth by 2035. Among the interventions included in ENAP is skin massage, which is considered a therapeutic-touch technique that can have positive effects on both the bodies and minds of infants [7]. However, despite the availability of evidence-based strategies for implementation, many nations have overlooked or inadequately implemented these strategies [8]. This highlights the need for improved efforts to ensure the effective and widespread adoption of interventions such as skin massage.

A systematic review conducted in 2013 found that emollient therapy has several benefits for preterm neonates, including enhanced weight gain, decreased risk of infection, and reduced newborn mortality. The review indicates that utilizing emollient therapy shows potential as an intervention in settings with limited resources [9].

The use of various types of emollient oils, such as sunflower oil, mustard oil, and vegetable oil, has gained popularity in recent years for skincare purposes. However, studies examining the efficacy of topical emollient oil application have yielded inconsistent results. Some studies have reported advantages of applying emollient oil on the skin of premature infants, including moisturization, reduction of water loss, and preservation of skin integrity [10–12], On the other hand, other studies have found no significant differences in weight gain when comparing

the use of emollient oil to no intervention [10,13,14]. It is worth noting that oil massage, including the use of emollient oils, is associated with potential side effects. These can include allergic reactions, rash, necrosis, and uremia [15,16].

Given the limited number of trials assessing the impact of emollient oil on preterm newborns, inconsistencies in the findings, and a lack of information regarding the appropriate dosage, frequency, duration, and selection of effective and safe emollient oils, it becomes crucial to address these inconsistencies, generate practical recommendations, and enhance the survival rates of preterm neonates through evidence-based interventions. Therefore, the objective of this study was to examine the effects of emollient oil applications on weight of preterm infants. The findings of this study can serve as valuable input for initiatives aimed at improving the survival outcomes of preterm newborns.

## Methods and materials

### Protocol registration

To ensure the avoidance of duplications, an initial search was conducted, encompassing narrative analysis and systematic reviews, along with registered protocols. The protocol for this systematic review and meta-analysis was registered in the International Prospective Register of Systematic Reviews (PROSPERO) under the reference number CRD42023413770. The methodology for the systematic review and meta-analysis was developed in accordance with the PRISMA (Preferred Reporting Items for Systematic Reviews and Meta-Analyses) reporting checklist, providing a comprehensive and standardized approach for reporting the findings (**S1 Appendix**).

### Eligibility criteria or inclusion criteria

The inclusion criteria for this study were

- **Study designs**: Randomized control trials studies

- **Publication status**: published articles or grey literatures.

- **Language**: articles that are written in English or those can be translated to English.

- **Study quality**: this meta-analysis included those studies with a Good, Fair and poor-quality score based on Cochrane Risk of Bias Tool for Randomized Controlled Trials criteria [17].

- **The outcome of interest**: all research articles that investigated the effect of emollient oil on preterm neonates' weights were included.

### Information sources

A comprehensive search was conducted on different databases, including PubMed, Cochrane, Scopus, Clinical trials, ProQuest Central, Epistemonikos, CINAHL, HINARI, Global Index Medicus and gray literature sources (Google scholar, Mednar, World cat, dissertation).

### Search strategy

Comprehensive searches were conducted in electronic databases and grey literature sources, encompassing all articles related to human subject research prior to the search date. The searches were conducted in the relevant search fields of electronic databases, employing sensitive search strategies that combined text words using Boolean operators and indexing terms

(PICO search) to ensure thorough coverage. The search period extended from July 7, 2023, to July 14, 2023 (**S2 Appendix**).

> 🔎 **PICO search guide**
>
> - ♣ Population–'Preterm newborn'
>
> - ♣ Intervention–'Emollient Oil application'
>
> - ♣ Comparison–'Standard care or massage without oil'
>
> - ♣ Outcomes–'weight'.

Additionally, cross-reference search was carried out to include other related studies from the final included studies that the database search might have missed.

## Study selection process

After conducting electronic database searches, the research articles were imported into End-note to identify and manage any duplicate publications. The article selection process for this review was carried out in three stages: initially based on titles alone, followed by abstracts, and finally full-text articles, in a sequential manner. To identify articles relevant to the topic of interest, the primary investigator (FSB) and coauthor (YA) independently screened the titles and abstracts using Rayan online software available at (https://www.rayansoftware.com). Any discrepancies that arose during this screening process were resolved through discussions between the investigators. Subsequently, a thorough review of the full-text articles was conducted to assess their eligibility based on predefined inclusion criteria. Specifically, studies investigating the impact of applying emollient oil on weight of preterm neonates were identified for further evaluation.

In cases where papers were not freely accessible, we contacted the corresponding authors via email to request access. Unfortunately, six articles were excluded from the final analysis because the corresponding authors did not respond to the emails (**S3 Appendix**).

**Data collection process (extraction.** A data extraction form was developed based on the guide from the Cochrane Collaboration for interventional reviews of randomized controlled trials (RCTs). This form served as a standardized tool to extract relevant data from the included studies. The data extraction was performed independently by Mr. FSB and Mr. AG, and subsequently reviewed by Mr. YA to enhance consistency and reliability. The data extraction form captured important details including the last name of the first author, publication year, place of study, frequency of application, dose and type of emollient oil, duration of intervention, sample size, and mean and standard deviation from both the intervention and control groups. To ensure efficient data management, a Microsoft Excel spreadsheet was utilized to record and organize the extracted data.

## Outcomes of the study

*Primary outcome*: Estimate the effect of emollient oil application in weight in preterm newborns (mean differences in grams). *Secondary outcome*: Assess adverse events (AEs) following applying topical emollient oil application in preterm newborns.

## Missing data management

We made an effort to address missing data in the articles by contacting the respective authors and requesting the relevant data points. However, in cases where it was not possible to obtain the missing data despite our attempts, we made the decision to exclude those particular papers

from the analysis. The reasons for their exclusion were documented and provided as part of the description in the analysis.

## Study risk of bias assessment

The quality evaluation of the included studies was conducted using the Cochrane Risk of Bias Tool for Randomized Controlled Trials (RoB 2.0). This tool consists of five domains that assess different sources of bias: bias arising from the randomization process, bias due to deviations from intended interventions, bias due to missing outcome data, bias in measurement of the outcome, and bias in selection of the reported result [18]. Each domain is rated as having a low risk of bias, high risk of bias, or some concerns.

The principal investigator, in collaboration with the coauthor (YA), independently rated each publication using the Cochrane RoB 2.0 tool. Once the quality evaluation was completed, the results were converted into the categories defined by the Agency for Healthcare Research and Quality (AHRQ) for easier interpretation. These AHRQ categories include Good, Fair, and Poor quality, which provide a summary assessment of the overall quality of the studies. (**S4 Appendix**).

## Data synthesis and analysis plan

Once the eligible articles with assured quality were identified, their findings were gathered, analyzed, and summarized. The STATA Version 17 software (STATA Corporation, College Station, Texas) was utilized to estimate the pooled effect estimates of emollient oil application on weight of premature babies.

Initially, a narrative overview of the selected articles was provided, highlighting key findings and trends. Subsequently, a summary table was created, which included details such as the author, year of publication, region, primary outcome, sample size, adverse events, quality score, and other significant results. This table served as a comprehensive reference for easy comparison and retrieval of information.

To facilitate comparisons of weight among preterm neonates, a mean difference (MD) along with a corresponding 95% confidence interval (CI) was employed as the effect measure. Forest plots, summary tables, and accompanying text were utilized to present a comprehensive summary of the results derived from the meta-analyses. These graphical and tabular representations provided a clear visualization of the overall findings and effect sizes across the analyzed studies.

## Heterogeneity exploration

To assess heterogeneity across the included studies, employed both graphical and statistical methods. Graphical methods included the use of forest plots and Galbraith plots, which provide visual representations of the variability in effect sizes across studies. Statistical tests were also utilized to assess heterogeneity. Cochran's Q test was employed to determine if the amount of heterogeneity between studies was greater than what would be expected by chance alone. A significance level of $P < 0.05$ was considered indicative of statistically significant heterogeneity. Additionally, the $I^2$ statistic was used to estimate the percentage of variability in results across studies that is due to real differences rather than chance. Threshold values of 25%, 50%, and 75% were considered as representing low, medium, and high levels of heterogeneity, respectively [19]. To further investigate and manage the causes of heterogeneity, random-effects model, subgroup analysis based on factors such as types of emollient oil, duration, dose, frequency of intervention, and study quality were employed. Sensitivity analysis and

meta-regression analysis were also conducted to explore potential sources of heterogeneity across studies.

## Small study effect diagnosis

Both graphical and objective tests were employed to evaluate the presence of publication bias. Funnel plots, a graphical method, were used to visually assess the symmetry of the distribution of study results. Asymmetry in the plot may indicate the presence of publication bias, specifically the small study effect. In addition to the graphical assessment, Egger's and Begg's tests to quantitatively examine publication bias. A statistically significant result (p-value < 0.05) in these tests suggests the presence of a small study effect, which can be indicative of publication bias [20]. Non-parametric trim and fill analysis employed to address the potential small study effect and publication bias. This method aims to estimate the potential number of missing studies due to publication bias and adjust the effect size accordingly. The analysis provides an adjusted effect size by imputing hypothetical missing studies to create symmetry in the funnel plot [21].

## Results

### Study selection

Out of the initial 2,734 articles retrieved, a total of 1,681 articles were excluded as they were identified as duplicates. Additionally, 995 articles were deemed unrelated to the goal of our study based on the screening of titles and abstracts. Consequently, the full texts of the remaining 58 publications were thoroughly reviewed to determine their eligibility. After this assessment, a total of 18 articles met the inclusion criteria and were included in the final analysis. These 18 articles were utilized to evaluate the impact of emollient oil application on weight of preterm newborns (**Fig 1**).

### Characteristics of included studies

This review included 18 randomized controlled trials (RCTs) conducted between 2000 and 2023, encompassing various continents. A total of 1454 preterm neonates participated in these trials, with 723 in the intervention group and 731 in the control group. The studies were conducted in nine different countries across five continents, including Asia [11,15,22–32], Australia [33], North and South America [34,35], and Europe [36,37]. The primary focus of these studies was to examine the effects of emollient oil therapy on weight in preterm infants.

Among the 18 included studies, 17 were conducted in institutional settings, while one study was community-based and carried out in a rural environment [15]. The total study population across the included studies ranged from 25 participants [36] to 258 participants [27]. Regarding the types of emollient oils used in the intervention groups, half of the studies employed coconut oil [15,26,27,38] or sunflower oil [22,29,31,32] as topical emollient therapies. The remaining studies utilized olive oil [25,28,30], Medium-chain triglyceride (MCT) oil [11,23,24], Soya oil [35], iSio4 [36,37], and Vimala [34] in their intervention groups. The duration of therapy for preterm neonates shows variation, spanning from 5 to 60 days, with an average duration of 18.5 days. However, we encountered difficulties in calculating the exact mean gestational age and baseline weight due to inconsistencies in reporting across the studies included in our analysis.

In the review, it is noted that six out of the eighteen included studies were published within the last five years. The remaining twelve studies were conducted between 2000 and December 2017, providing a range of data from different time periods. To summarize and draw

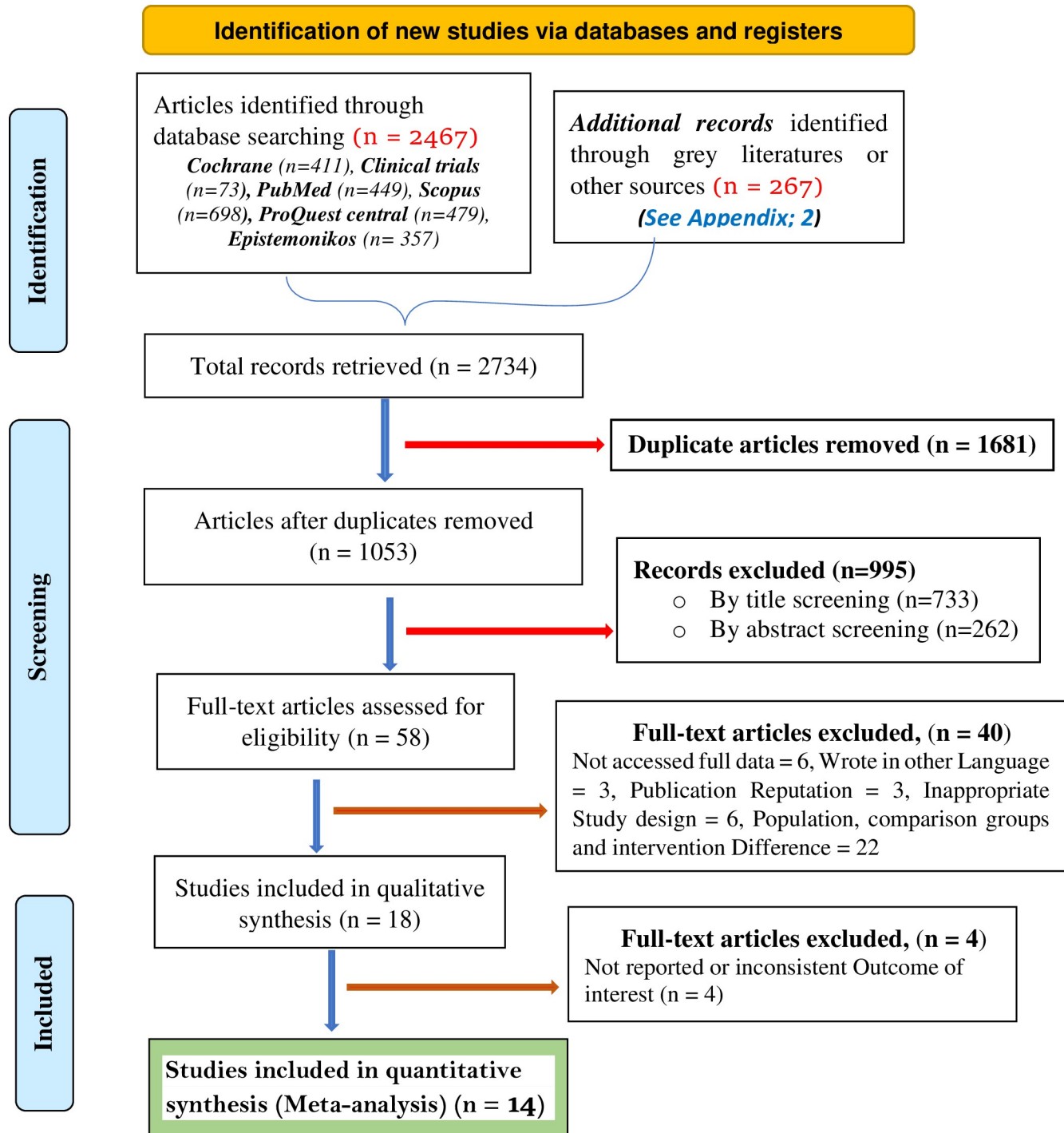

**Fig 1. PRISMA flow diagram for the selection of eligible studies on effect of topical emollient oil application on weight gain of preterm neonate, 2023.**

conclusions regarding the effect of topical emollient oil application on weight of preterm newborns, a narrative synthesis was performed. This synthesis involved extracting key findings from the full-text articles and summarizing them (Table 1).

**Table 1.** *Characteristics of included studies* showed pooled difference in mean weight on effect of topical emollient oil application on weight gain of preterm neonate, 2023.

| First Author name, Year | Country | Study setting | Study population | Sample size Intervention | Sample size control | Total participant | Frequency (# times/ day) | Dose (ml/ kg/ day) | Duration (Days) | Emollients Type | Comparison group | Primary Outcome parameters | Reported Adverse event | Quality score (AHRQ Standards) |
|---|---|---|---|---|---|---|---|---|---|---|---|---|---|---|
| Jabraeile M, 2016 [28] | Iran | NICU | 28 and 32 weeks | 42 | 44 | 86 | 3 | NR | 10 | olive oil | massage without oil | weight gain | NR | Fair |
| Arora J, 2005 [32] | India | NICU | less than 37 weeks | 20 | 23 | 43 | 4 | 10 | 28 | Sunflower Oil | massage without oil | weight gain | NR | Fair |
| K. Sankaranarayanan, 2005 [26] | India | NICU | preterm | 38 | 37 | 75 | 4 | NR | 29 | Coconut oil | routine skin care | weight gain | Mild rash | Fair |
| Strunk T, 2017 [33] | Australia | NICU | <30 weeks | 36 | 36 | 72 | 2 | 5 | 21 | Coconut oil | routine skin care | neonatal skin condition | No adverse events | Fair |
| Al-Abdullah, 2012 [25] | Saudi Arabia | NICU | ≤ 34 weeks | 26 | 25 | 51 | 2 | 1.5 | 14 | Olive oil | routine skin care | Weight & Infection prevention | NR | Fair |
| Khatun N, 2021 [15] | India | Community | preterm | 119 | 128 | 247 | 4 | 5 | 30 | Coconut oil | massage without oil | weight gain | accidental slippage | Good |
| Liao Y, 2021 [11] | Taiwan | NICU | 28 and 37weeks | 16 | 16 | 32 | 3 | 10 | 7 | MCT | massage without oil | Weight gain | No adverse events | Good |
| Oriot D, 2008 [36] | France | NICU | 31- to 34-week | 12 | 13 | 25 | 2 | 5 | 10 | oil ISIO4 | routine skin care | weight gain | No adverse events | Good |
| Gonzalez AP, 2009 [34] | Mexico | NICU | 30 to 35 weeks | 30 | 30 | 60 | 2 | NR | 10 | Vimala | routine skin care | weight gain | No adverse events | Good |
| Montaseri S, 2020 [30] | Iran | NICU | 30th to 36th weeks | 15 | 15 | 30 | 2 | NR | 5 | olive oil | routine skin care | weight gain | No adverse events | Good |
| Fallah, Razieh, 2013 [31] | Iran | NICU | 33–37 weeks | 27 | 27 | 54 | 3 | 10 | 14 | sunflower oil | massage without oil | weight | No adverse events | Good |
| Salam RA, 2015 [27] | Pakistan | NICU | 26–37 weeks | 128 | 130 | 258 | 2 | 5 | 28 | Coconut oil | routine skin care | Infection prevention & weight | No adverse events | Good |
| Soriano R, 2000 [35] | Brazil | NICU | 28–34 wks, | 29 | 31 | 60 | 3 | 12 | 30 | Soybean Oil | routine skin care | Weight gain | NR | Poor |
| Armand M, 2022 [37] | France | NICU | very premature | 18 | 18 | 36 | 1 | 10 | 5 | oil ISIO4 | massage without oil | weight gain | NR | Fair |
| Saeadi, Reza, 2015 [23] | Iran | NICU | < 37 weeks | 23 | 24 | 47 | 4 | 10 | 7 | MCT | massage without oil | weight gain | NR | Poor |
| Jamshaid AA, 2021 [22] | Pakistan | NICU | 28 and 37 weeks | 70 | 70 | 140 | 2 | 10 | 60 | Sunflower oil | massage without oil | weight and length | NR | Poor |
| Kumar and Upadhyay, 2013 [29] | India | NICU | < 35 weeks & < 1800 gram | 25 | 23 | 48 | 4 | 10 | 28 | Sunflower oil | routine skin care | Weight gain | NR | Fair |

(*Continued*)

**Table 1.** (Continued)

| First Author name, Year | Country | Study setting | Study population | Sample size Intervention | Sample size control | Total participant | Frequency (# times/ day) | Dose (ml/ kg/ day) | Duration (Days) | Emollients Type | Comparison group | Primary Outcome parameters | Reported Adverse event | Quality score (AHRQ Standards) |
|---|---|---|---|---|---|---|---|---|---|---|---|---|---|---|
| Saeidi, 2009 [24] | Iran | NICU | preterm | 40 | 41 | 81 | 4 | 10 | 7 | MCT | routine skin care | weight gain | NR | Fair |

*:- **NICU:** *Neonatal intensive care unit*; **MCT:** *Medium chain triglycerides*; **HC:** *Head circumference*; **HAI:** *Hospital acquired infection*; **NR:** *Not reported.*

In terms of adverse events related to topical emollient oil application, a total of nine studies investigated this aspect. Out of these, only two studies reported specific adverse events in the intervention groups, namely rash and accidental slippage [15,26].

## Effect of emollient oil on preterm weight

In the meta-analysis, the pooled mean difference was calculated based on data from 14 studies, involving a total of 1067 preterm neonates. Among these, 531 neonates were in the treatment group receiving topical emollient oil, while 536 neonates were in the control group.

The mean difference in weight gain varied across the included studies, ranging from 8.34 grams [25] to 313.2 gram [30]. To obtain a summary estimate of the effect of topical emollient oil application on weight of preterm babies, the pooled mean difference was calculated using fixed and random effects models.

However, because to the notable heterogeneity among the included studies ($I^2 > 93.24\%$, p 0:000), the final values were reported using a random effect model. The pooled mean difference was found to be 52.15 grams (95% CI: 45.96, 58.35) by using the DerSimonian and Laird random-effects model (**Fig 2**). The blue square in the figure represents the mean difference for each study according to the random-effects model, while the length of the segment it lies on represents the confidence interval for each study. The diamond symbol represents the overall combined mean difference for all the studies, which indicates the pooled estimate of the effect size.

**Subgroup analysis effect of emollient oil on preterm weight.** Subgroup analyses were conducted in response to the observed higher levels of heterogeneity among the included studies, which were evident both subjectively (forest plot and Galbraith plot) and quantitatively ($I^2$

**Effect of emollient oil on preterm weight gain**

| Study | N | Treatment Mean | SD | N | Control Mean | SD | Mean diff. with 95% CI | | | Weight (%) |
|---|---|---|---|---|---|---|---|---|---|---|
| Jabraeile et al., 2016 | 42 | 211.11 | 102 | 44 | 72.61 | 114.86 | 138.50 [ | 92.51, | 184.49] | 1.68 |
| Arora et al., 2005 | 20 | 365.8 | 165.2 | 23 | 285 | 170.4 | 80.80 [ | -19.88, | 181.48] | 0.37 |
| K. Sankaranarayanan et al., 2005 | 38 | 197.11 | 46.86 | 37 | 138.27 | 45.11 | 58.84 [ | 38.01, | 79.67] | 6.31 |
| Al-Abdullah et al., 2012 | 26 | 251.54 | 12.47 | 25 | 243.2 | 18.42 | 8.34 [ | -0.26, | 16.94] | 15.48 |
| Armand et al., 2022 | 18 | 128 | 77 | 18 | 110 | 46 | 18.00 [ | -23.44, | 59.44] | 2.03 |
| Saeidi et al., 2009 | 40 | 105 | 3.1 | 41 | 54 | 3.1 | 51.00 [ | 49.65, | 52.35] | 21.85 |
| Khatun et al., 2021 | 119 | 160.3 | 30.6 | 128 | 110 | 20.9 | 50.30 [ | 43.80, | 56.80] | 17.76 |
| Gonzalez et al., 2009 | 30 | 188.2 | 41.2 | 30 | 146.7 | 56.43 | 41.50 [ | 16.50, | 66.50] | 4.80 |
| Fallah et al., 2013 | 27 | 216 | 44 | 27 | 90 | 99 | 126.00 [ | 85.14, | 166.86] | 2.08 |
| Montaseri et al., 2020 | 15 | 460 | 418.69 | 15 | 146.8 | 327.37 | 313.20 [ | 44.24, | 582.16] | 0.05 |
| Soriano R, 2000 | 29 | 703 | 129 | 31 | 576 | 140 | 127.00 [ | 58.74, | 195.26] | 0.79 |
| Saeadi, Reza, 2015 | 32 | 105 | 1.3 | 24 | 52 | .1 | 53.00 [ | 52.48, | 53.52] | 22.05 |
| Jamshaid AA, 2021 | 70 | 489.84 | 297.48 | 70 | 373.43 | 276.31 | 116.41 [ | 21.30, | 211.52] | 0.42 |
| Kumar and Upadhyay, 2013 | 25 | 476.76 | 47.9 | 23 | 334.96 | 46.4 | 141.80 [ | 115.08, | 168.52] | 4.32 |
| **Overall** | | | | | | | **52.15 [** | **45.96,** | **58.35]** | |

Heterogeneity: $\tau^2 = 45.22$, $I^2 = 93.24\%$, $H^2 = 14.80$
Test of $\theta_i = \theta_j$: Q(13) = 192.38, p = 0.00
Test of $\theta = 0$: z = 16.51, p = 0.00

Random-effects DerSimonian–Laird model

**Fig 2. Forest plot showing polled mean weight difference effect of topical emollient oil application on weight gain of preterm neonate, 2023.**

and Q test). The type of emollient oil used, the duration of therapy, dose of therapy, frequency of therapy, studies quality score was taken into consideration when doing these analyses.

Subgroup analyses revealed improvements in heterogeneity when considering types of emollient oil used, the dose of therapy, and the duration of therapy. In terms of the types of emollient oil, preterm newborns who received massages with coconut oil showed a mean difference in weight gain of 51.06 grams (95% CI: 44.86, 57.26). On the other hand, infants who received massages with sunflower oil had a higher mean difference of 133.52 grams (95% CI: 112.24, 154.8) compared to their counterparts. Regarding the dose of therapy, infants who received emollient oil massages three times a day exhibited a significant higher mean weight increment of 130.76 grams (95% CI: 102.88, 158.64) compared to other dosing frequencies (Table 2).

**Meta regression.** Meta-regression analysis was conducted to determine whether specific covariates could account for some of the heterogeneity observed. The results of the meta-regression analysis indicated that the type of emollient oil used, the duration of therapy, and the frequency of therapy collectively explained a significant proportion of the observed heterogeneity.

Specifically, the emollient oil type accounted for 36.6% of the heterogeneity, the duration of therapy explained 34.39% of the heterogeneity, and the frequency of therapy explained 59.85% of the heterogeneity in the bi-variable regression analysis. On the other hand, variables such as study setting and study quality, as well as multiple regression of covariates, did not significantly influence the heterogeneity observed among the included studies (Table 3).

**Sensitivity analysis of mean weight difference of premature babies.** To evaluate the influence of individual studies on the overall meta-analysis estimate, sensitivity analyses or influential analyses were conducted. These analyses aimed to assess the impact of removing outlier studies from the analysis.

Based on the sensitivity analysis, it was identified that two studies were considered outliers or influential papers. This determination was made by examining the pooled estimate after

**Table 2. Subgroup analysis showed pooled difference in mean weight on effect of topical emollient oil application on weight gain of preterm neonate, 2023.**

| Subgroup's variables | Category | No of Study | Mean difference | 95% CI | $I^2$ (%) | P-value |
|---|---|---|---|---|---|---|
| Emollient oil type | Coconut oil | 2 | **51.06** | **[44.86, 57.26]** | **0.00** | **0.44** * |
| | Olive oil | 3 | 106.08 | [-17.52, 229.68] | 94.21 | 0.00 |
| | Sunflower oil | 4 | **133.52** | **[112.24, 154.8]** | **0.00** | **0.64** * |
| | Other* | 5 | 51.93 | [49.42, 54.45] | 73.95 | 0.00 |
| Duration of therapy | Before 15 days | 8 | 44.94 | [38.20, 51.68] | 95.08 | 0.00 |
| | 15 days or above | 6 | 91.32 | [53.57, 129.07] | 89.70 | 0.00 |
| Dose of therapy | 5 ml/kg or less | 2 | 29.42 | [- 11.7, 70.54] | 98.28 | 0.00 |
| | 10 ml/kg | 8 | 57.13 | [51.55, 62.71] | 90.22 | 0.00 |
| Frequency of therapy | 2 times/day | 5 | 34.44 | [2.10, 66.77] | 74.20 | 0.00 |
| | 3 times/day | 3 | **130.76** | **[102.88, 158.64]** | **0.00** | **0.92** * |
| | 4 times/day | 6 | 54.04 | [49.86, 58.21] | 90.23 | 0.00 |
| Study Quality | Fair | 6 | 51.77 | [24.77, 78.77] | 95.44 | 0.00 |
| | Good | 5 | 94.25 | [47.36, 141.15] | 93.18 | 0.00 |
| | Poor | 3 | 87.79 | [30.81, 144.76] | 67.86 | 0.04 |

Other

* (MCT, Isio4 and Vimila).

**Table 3. Meta-regression showed pooled difference in mean weight on effect of topical emollient oil application on weight gain of preterm neonate, 2023.**

| Variables | No of Studies | Coefficient | [95% confidence] | | I² (%) | R-squared (%) | P- value |
|---|---|---|---|---|---|---|---|
| | | | Lower | Upper | | | |
| Type of emollient oil | 13 | -2.82 | -4.93 | -0.71 | 88.62 | **36.6** | **0.013** |
| Duration of therapy | 13 | 3.71 | 1.03 | 6.41 | 88.75 | **34.39** | **0.011** |
| Frequency of therapy | 13 | 11.4 | 7.37 | 15.57 | 83.06 | **59.85** | **0.000** |
| Study setting | 13 | -3.15 | -19.9 | 13.6 | 92.62 | 0.00 | 0.687 |
| Study quality | 13 | 13.39 | -8.46 | 35.26 | 91.74 | 0.00 | 0.204 |
| *Multi-Variable* | All five variables at once | | | | 79.16 | 0.00 | 0.166 |

excluding the individual study's point estimate, which resulted in the estimate falling outside the confidence interval of the "combined" analysis (95% CI: 45.96, 58.35) (**Fig 3**).

The sensitivity analysis identified two specific studies, Saeidi et al. (2009) and Saeidi, Reza (2015), as having a notable impact on the overall estimation in the meta-analysis. The original pooled difference in mean weight gain was estimated to be 52.15 grams (95% CI: 45.96, 58.35). However, after excluding the outlier studies, the pooled difference in mean weight changed to 78.57grams (95% CI: 52.46, 104.68) (**Table 4**).

## Sensitivity Analysis of mean weight difference of premature babies

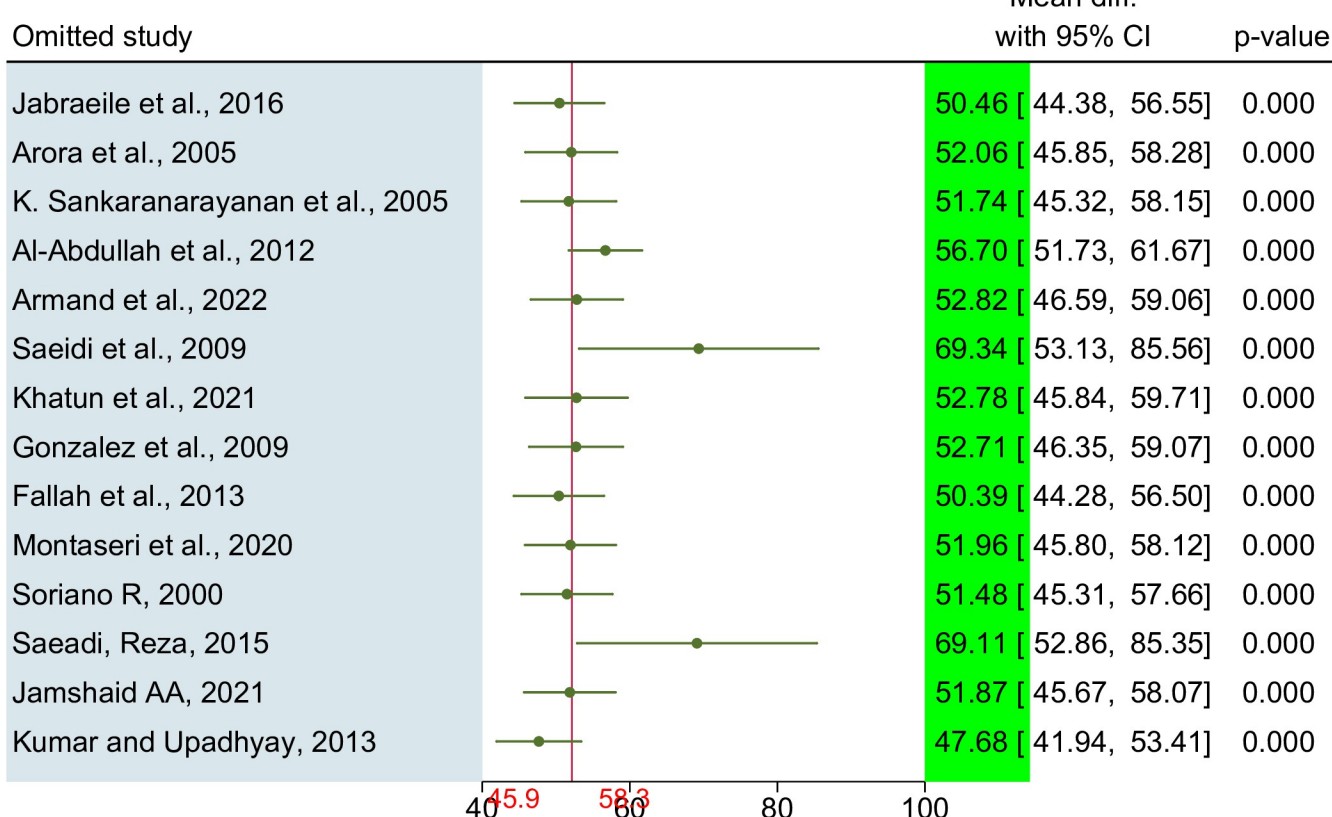

**Fig 3. Sensitivity Analysis showing polled mean weight difference effect of topical emollient oil application on weight gain of preterm neonate, 2023.**

**Table 4. Pooled difference in mean weight effect of topical emollient oil application on weight gain of preterm neonate, 2023.**

| S. no | Study | Mean diff. | [95% conf. | | Weight (%) |
|---|---|---|---|---|---|
| 1 | Jabraeile et al., 2016 [28] | 138.5 | 92.509 | 184.491 | 8.74 |
| 2 | Arora et al., 2005 [32] | 80.8 | -19.879 | 181.479 | 4.31 |
| 3 | K. Sankaranarayanan, 2005 [26] | 58.84 | 38.015 | 79.665 | 11.15 |
| 4 | Al-Abdullah et al., 2012 [25] | 8.34 | -0.262 | 16.942 | 11.84 |
| 5 | Armand et al., 2022 [37] | 18 | -23.436 | 59.436 | 9.21 |
| 6 | Khatun et al., 2021 [15] | 50.3 | 43.804 | 56.796 | 11.91 |
| 7 | Gonzalez et al., 2009 [34] | 41.5 | 16.498 | 66.502 | 10.81 |
| 8 | Fallah et al., 2013 [31] | 126 | 85.136 | 166.864 | 9.27 |
| 9 | Montaseri et al., 2020 [30] | 313.2 | 44.238 | 582.162 | 0.87 |
| 10 | Soriano R, 2000 [35] | 127 | 58.744 | 195.256 | 6.59 |
| 11 | Jamshaid AA, 2021 [22] | 116.41 | 21.299 | 211.521 | 4.63 |
| 12 | Kumar and Upadhyay, 2013 [29] | 141.8 | 115.078 | 168.522 | 10.66 |
| | *Overall estimation* | *78.57* | *52.46* | *104.68* | |
| | *I2 and Q statistics* | 93.46% with P-value = 0.000 | | | |

**Publication bias assessmen.** To assess the potential presence of publication bias, two methods were employed: funnel plots and Egger's statistical test. The funnel plot, a graphical test, revealed a slight dispersion of studies, particularly on the right side, indicating the presence of potential publication bias. This suggests that there may be a tendency for small studies with positive outcomes to be published (**Fig 4**). Additionally, Egger's test was conducted to statistically evaluate the presence of small study effects (p-value = 0.000).

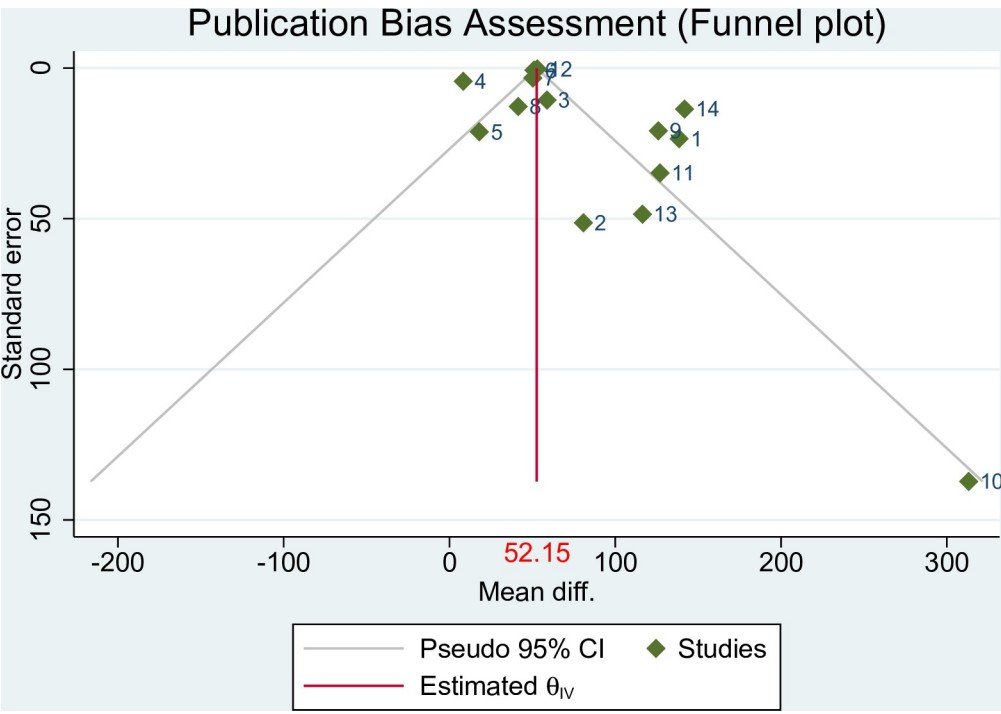

**Fig 4. Funnel plot showed polled mean weight difference on effect of topical emollient oil application on weight gain of preterm neonate, 2023.**

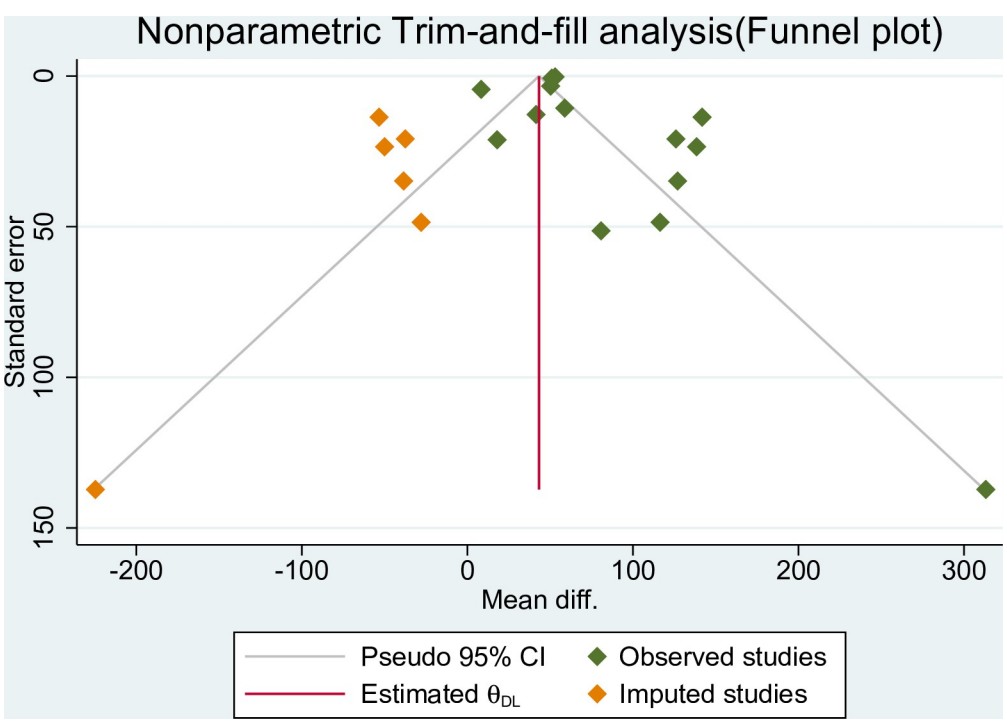

**Fig 5. Nonparametric Trim and fill analysis showed polled mean weight difference on effect of topical emollient oil application on weight gain of preterm neonate, 2023.**

To address the potential impact of publication bias and adjust for its influence, a nonparametric Trim-and-fill analysis was performed. The analysis revealed a significant change in the pooled mean difference, shifting from 52.15 grams to 43.3 grams (95% CI: 36.39, 50.22), after imputing six studies using the random effects model (**Fig 5**).

## Discussion

In this systematic review and meta-analysis, the primary objective was to assess the effect of emollient oil application on weight gain in preterm neonates. The analysis considered studies conducted in various countries at the global level in order to capture a wide range of evidence. After extensive searching and quality assessment process, a total of eighteen studies were identified for the estimation of weight gain in preterm neonates. However, after further screening and applying inclusion criteria, fourteen studies were included in the final analysis of this meta-analysis. These fourteen studies met the specified criteria and provided relevant data for the analysis of the effect of emollient oil application on weight gain in preterm neonates.

The study's findings indicate that the pooled difference in mean weight among preterm neonates who received emollient therapy was estimated to be 52.15 grams (95% CI: 45.96, 58.35). This result aligns with evidence generated from developing countries, suggesting a consistent effect of emollient therapy on promoting postnatal growth in preterm infants [9]. Furthermore, recent emerging evidence also supports the notion that emollient therapy contributes to enhanced postnatal growth [9,39].

Considering the significance of optimal growth and its potential impact on the short- and long-term health outcomes of preterm neonates, a weight gain of 52 grams, although seemingly modest, can be considered clinically relevant. This weight gain has the potential to contribute to improved organ maturation, enhanced thermoregulation, and a decreased risk of

complications such as hypoglycemia and respiratory distress syndrome [40,41]. Additionally, it can help reduce the likelihood of long-term growth faltering and associated health complications [42].

However, it is important to consider the results of the nonparametric trim and fill analysis, which revealed a decline in the pooled difference in mean weight to 43.3 grams (95% CI: 36.39, 50.22). This adjustment may be attributed to publication bias, suggesting that studies with small sample sizes and negative results may be underrepresented in the published literature.

The study further suggests that preterm infants who received massages with sunflower oil and coconut oil exhibited better weight gain compared to control groups. This finding could be explained by the presence of fatty acids, such as linoleic acid, and vitamin D in these oils. These components may enhance the binding to peroxisome proliferator–activated receptor α receptors on keratinocytes, facilitating skin development, maturation, and potentially preventing complications such as hypothermia and apnea [12,37,43].

Another significant finding of the study reveals that preterm neonates who received massages three times per day exhibited a notable increase in weight compared to their counterparts. This effect could be attributed to the impact of massage or tactile stimulation on the physiological stability and development of preterm newborns [44]. The mechanical stimulation provided by oil massage may stimulate the production of proteins and the growth hormone IGF-1, ultimately leading to weight gain [45]. Moreover, regular oil massage has the potential to enhance blood flow, regulate body temperature, and reduce the energy required to maintain body temperature, thereby supporting healthier weight gain [9]. Additionally, oil massage improves the skin's barrier function, reducing trans-epidermal water loss and facilitating weight gain in premature newborns [46].

Furthermore, the study suggests that the observed weight gain in preterm neonates through oil massage may contribute to a reduced hospital stay. Consistent weight gain is often one of the discharge criteria from the Neonatal Intensive Care Unit (NICU). Therefore, the positive effect of oil massage on weight gain may expedite the hospital discharge process for premature babies.

In conclusion, the moderate-to-high level of certainty regarding the effectiveness of topical emollient oil application in promoting weight gain among preterm neonates provides strong support for its use as a beneficial intervention. This finding indicates a positive impact on weight gain outcomes in this population, highlighting the reliability and confidence in this conclusion.

## Limitations of the study

The study encountered notable limitations that affected its conclusions. A primary concern was the scarcity of available primary studies, which restricted the researchers' ability to estimate the mean difference across different continents and skin colors. The absence of data from Africa and the heterogeneous nature of the available data were also limited the generalizability of the findings. Another important limitation was the insufficient discussion on publication bias and other influential factors, including nutrition and age at enrollment, which could impact weight gain. It is important to consider that weight gain in preterm neonates is a multifactorial event, not solely attributable to body massage with emollient oil. Additionally, the study did not adequately explore the challenges associated with weight gain during the first week of life in preterm neonates, which is a critical period. These aspects should be further explored and discussed in future studies to provide a more comprehensive understanding of the factors influencing weight gain in this population.

## Conclusion

Considering the significant global public health concern surrounding premature birth and its associated mortality, it is crucial to implement low-tech and cost-effective interventions to reduce preterm newborn deaths. This systematic review and meta-analysis indicate that the application of sunflower oil and coconut oil during massage resulted in better weight gain for preterm neonates compared to those who did not receive oil massage or only received routine care.

Furthermore, preterm babies who received massage three times a day demonstrated improved weight gain compared to control groups. Additionally, the limited reporting of adverse events highlights the need for further investigation into potential risks associated with emollient oil application.

The significant pooled difference in mean weight gain observed indicates that preterm infants who received emollient oil experienced improved nutrition, reduced infection risk, and potentially better neurodevelopmental outcomes. Therefore, it is essential for local policy-makers and health planners to prioritize the implementation of emollient oil use in routine care for preterm infants. Integrating this evidence into national guidelines for NICU settings would greatly contribute to facilitating weight gain and overall well-being.

## Supporting information

**S1 Appendix. PRISMA checklist.**
(PDF)

**S2 Appendix. Search strategies queries results.**
(PDF)

**S3 Appendix. List of excluded studies from final analysis.**
(PDF)

**S4 Appendix. Quality assessment based on cochrane risk of bias tool for RCTs 'ROB 2'.**
(PDF)

## Acknowledgments

We are thankful to the authors of the original studies included in this systematic review and meta-analysis, and to those who contributed a lot in this work.

## Author Contributions

**Conceptualization:** Fekadeselassie Belege Getaneh, Alemu Gedefie Belete, Yibeltal Asmamaw, Zemen Mengesha, Zebenay Workneh Bitew, Asressie Molla.

**Data curation:** Anissa Mohammed, Amare Muche, Aznamariyam Ayres, Yibeltal Asmamaw, Asrat Dimtse, Dires Birhanu Mihretie, Meaza Mengstu.

**Formal analysis:** Fekadeselassie Belege Getaneh, Amare Muche, Yibeltal Asmamaw, Natnael Moges Misganaw, Zebenay Workneh Bitew, Asressie Molla.

**Methodology:** Fekadeselassie Belege Getaneh, Anissa Mohammed, Alemu Gedefie Belete, Amare Muche, Aznamariyam Ayres, Yibeltal Asmamaw, Zemen Mengesha, Asrat Dimtse, Dires Birhanu Mihretie, Zebenay Workneh Bitew, Meaza Mengstu.

**Software:** Fekadeselassie Belege Getaneh, Aznamariyam Ayres, Yibeltal Asmamaw, Natnael Moges Misganaw, Zebenay Workneh Bitew, Asressie Molla.

**Visualization:** Alemu Gedefie Belete.

**Writing – original draft:** Fekadeselassie Belege Getaneh, Alemu Gedefie Belete, Yibeltal Asmamaw, Asrat Dimtse, Dires Birhanu Mihretie, Asressie Molla.

**Writing – review & editing:** Anissa Mohammed, Amare Muche, Aznamariyam Ayres, Zemen Mengesha, Natnael Moges Misganaw, Zebenay Workneh Bitew, Meaza Mengstu, Asressie Molla.

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
