## [Decision Letter · Decision Letter 0]

11 Mar 2024

PONE-D-24-05779EFFECT OF TOPICAL EMOLLIENT OIL APPLICATION ON WEIGHT GAIN OF PRETERM NEWBORNS: A SYSTEMATIC REVIEW AND META-ANALYSISPLOS ONE

Dear Dr. Getaneh,

Thank you for submitting your manuscript to PLOS ONE. After careful consideration, we feel that it has merit but does not fully meet PLOS ONE’s publication criteria as it currently stands. Therefore, we invite you to submit a revised version of the manuscript that addresses the points raised during the review process.

The manuscript under review is a critical and relevant systematic review, however the reviewers have highlighted some major concerns and I concur. I would suggest that the authors carefully respond to all the concerns raised by the reviewers (especially around missing studies) and revise the manuscript as suggested. I would also encourage the authors to give special consideration to the langauge and editing when resubmitting.

We look forward to receiving your revised manuscript.

Kind regards,

Rehana Abdus Salam

Academic Editor

PLOS ONE

Journal Requirements:

2. We note that you have referenced (unpublished) on page 13, which has currently not yet been accepted for publication. Please remove this from your References and amend this to state in the body of your manuscript: (ie “Bewick et al. [Unpublished]”) as detailed online in our guide for authors

Additional Editor Comments:

The manuscript under review is a critical and relevant systematic review, however the reviewers have highlighted some major concerns and I concur. I would suggest that the authors carefully respond to all the concerns raised by the reviewers (especially around missing studies) and revise the manuscript as suggested. I would also encourage the authors to give special consideration to the langauge and editing when resubmitting.

Reviewers' comments:

Reviewer's Responses to Questions

**Comments to the Author**

1. Is the manuscript technically sound, and do the data support the conclusions?

Reviewer #1: Partly

Reviewer #2: Yes

2. Has the statistical analysis been performed appropriately and rigorously? 

Reviewer #1: I Don't Know

Reviewer #2: Yes

3. Have the authors made all data underlying the findings in their manuscript fully available?

Reviewer #1: Yes

Reviewer #2: Yes

4. Is the manuscript presented in an intelligible fashion and written in standard English?

Reviewer #1: Yes

Reviewer #2: No

5. Review Comments to the Author

Reviewer #1: In this systematic review and meta-analysis, the authors reviewed the existing literature on the efficacy of topical applications of emollient application on weight gain in pre-term newborns. The application of topical emollients has been showing several benefits on preterm newborns and severely malnourished children in terms of weight gain, protection from infections, increased fatty acid levels, and reduced mortality. This systematic review and meta-analysis would generate substantial evidence on the efficacy of emollients on pre-term newborns' weight gain from global studies. This study followed a standard methodological technique to do a systematic review and meta-analysis. It is also well written, and the results are well presented. However, the authors need to consider the following suggestions to revise this manuscript to make it compatible for publication.

Comments:

1. There was a published systematic review by Rehana AS et al., 2013 ( DOI: 10.1186/1471-2458-13-S3-S31) that mentioned the efficacy of emollient therapy on newborns’ weight gain. The authors of this study should justify how this review would generate additional evidence and value over the previous article by Rehana AS et al.

2. The authors mentioned that the rationale of this study is to synthesise current evidence on interventions to improve survival outcomes in preterm infants, and emollient therapy would be one of the important management strategies. However, based on existing evidence of the beneficial effects of emollient therapy, the WHO has already recommended the application of ‘natural’ oils (e.g. sunflower and coconut oil) to the body of preterm or LBW infants to increase weight gain ( DOI: 10.1016/j.eclinm.2023.102155). Priority for further research has been set to “Which emollients (which oils, which composition) are most effective and safe? And their optimal regime (dose, frequency, duration) and mode of application” (DOI: 10.1016/j.eclinm.2023.102126). Thus, the authors should describe strong arguments on how the findings of this review study add further value to the scientific community.

3. The authors concluded with that “Therefore, Policymakers and health planners should

give great emphasis on implementation of the use of emollient oils in the routine care of preterm infants to assist with weight gain..” As per my previous comment, the policy makers ( e.g. WHO) has already considered and recommended emollient therapy in to a standard management therapy. The author should mention what further benefit can be achieved from this study.

4. The study included 17 articles published between 2000 to July 2023. But I wonder if they missed more important RCTs on this topic that were published in this time frame. For example, Kumar V et al, 2022 (doi: 10.1093/ajcn/nqab430.); Sankaranarayanan K, et al., 2005, PMID: 16208048; Kumar J et al, 2012 (DOI: 10.1007/s12098-012-0869-7). Authors should revise their search and include all missed articles.

5. The authors did the pooled analysis on the mean weight gain after emollient therapy, and it showed significant results. However, their studies were highly heterogeneous. They differ in terms of emollient type, duration and dose. Thus, it may reduce the strength of the study results. Although the authors did subgroup analysis on the pooled analysis, the sample size is too low to strengthen the results. For example, the coconut oil group had two studies, 5ml/kg/day oil group had two studies.

6. The authors did a quality evaluation of the studies by using the Cochrane Risk of Bias Tool for Randomized Controlled Trials (Higgins et al., 2011), which is an old tool. The ‘RoB 2’ has been revised in August 2019. So, the author should use the updated tool for a better acceptability of their methodology.

7. As the authors mentioned, the weight gain is multifactorial. Yes, it depends on the combination of the type of emollient oil, its dose and duration. Pooled analysis on homogenous methodological studies can further refute the findings of this study. Thus, research priority has been set to assess the efficacy of the best emollient oil across all available oils for a specific dose and duration. The authors should describe this point, describe the weakness of the heterogeneous study analysis, and come up with some recommendations for future RCTs.

Reviewer #2: The authors of ‘Effect Of Topical Emollient Oil Application On Weight Gain Of Preterm Newborns: A Systematic Review And Meta-Analysis’ have worked on a relevant topic and the review is well performed but my major concern is that this would need thorough language correction and editing.

Other comments

The background does not cite all the previous systematic reviews that have been done for this intervention looking at weight and also other outcomes.

Rationale for this systematic review needs to be strengthened

Methods should start with objectives

The authors should mention that how many authors were contacted for full texts and missing data and how many responded

Was ROB 1 used or ROB 2 used for quality assessment.

Methods should define the planned sub-groups

What was the time or age at which weight was measured.

For conclusion, the authors need to make the clinical relevance of about 50g gain in weight and also mention that over what time this change occurred.

6. PLOS authors have the option to publish the peer review history of their article (what does this mean?). If published, this will include your full peer review and any attached files.

Reviewer #1: No

Reviewer #2: No

---

## [Author Response · Author response to Decision Letter 0]

29 Mar 2024

Dear Editors and Reviewers,

We would like to express our sincere appreciation for your dedicated effort in critically reviewing our work. Your valuable feedback has played a pivotal role in enhancing the quality of our manuscript and ensuring the robustness and significance of the study's findings.

Responses for Reviewer One

1. There was a published systematic review by Rehana AS et al., 2013 (DOI: 10.1186/1471-2458-13-S3-S31) that mentioned the efficacy of emollient therapy on newborns’ weight gain. The authors of this study should justify how this review would generate additional evidence and value over the previous article by Rehana AS et al.

Response: Thank you for your valuable suggestions. We acknowledge the previous study conducted by Rehana AS et al. and appreciate their contributions to the field. Their study focused specifically on research conducted in the developing world and included data up until December 2012. Furthermore, their analysis was limited to four studies with a sample size of 300, primarily examining the weight difference among groups.

In contrast, our study aims to build upon their research by providing an updated and comprehensive analysis of the impact of emollient oil application on weight gain in preterm newborns. We have expanded the scope and timeframe of our analysis to include studies conducted worldwide until 2023. This broader approach has allowed us to incorporate a larger sample size of 1,406 participants, enabling us to capture a more diverse range of studies and enhance the statistical power of our findings.

2. The authors mentioned that the rationale of this study is to synthesise current evidence on interventions to improve survival outcomes in preterm infants, and emollient therapy would be one of the important management strategies. However, based on existing evidence of the beneficial effects of emollient therapy, the WHO has already recommended the application of ‘natural’ oils (e.g. sunflower and coconut oil) to the body of preterm or LBW infants to increase weight gain ( DOI: 10.1016/j.eclinm.2023.102155). Priority for further research has been set to “Which emollients (which oils, which composition) are most effective and safe? And their optimal regime (dose, frequency, duration) and mode of application” (DOI: 0.1016/j.eclinm.2023.102126). Thus, the authors should describe strong arguments on how the findings of this review study add further value to the scientific community.

Response: Thank you for your concerns and suggestions. Despite the recommendation by the World Health Organization (WHO) regarding the application of emollient oil in preterm and low birth weight babies, there still exists a controversy among the results reported in the literature. Our study aims to address these controversies and provide valuable insights into the use of emollient oils in this population.

Moreover, our study goes beyond examining the overall impact of emollient oil application. We also investigate important factors such as the specific type of emollient oil used, the frequency and dosage of application, as well as potential adverse events associated with its use. By addressing these aspects, our research contributes to enhancing the practical application of this intervention in day-to-day clinical practice.

Furthermore, another crucial aspect of our study is the identification of research gaps and areas of uncertainty. By highlighting these gaps, we aim to guide and inspire further research in this field. This comprehensive approach not only provides valuable insights for clinical decision-making but also serves as a foundation for future investigations.

We have made a concerted effort to include these significant points in the main document, ensuring that the implications of our study are clearly communicated.

3. The authors concluded with that “Therefore, Policymakers and health planners should give great emphasis on implementation of the use of emollient oils in the routine care of preterm infants to assist with weight gain.” As per my previous comment, the policy makers ( e.g. WHO) has already considered and recommended emollient therapy in to a standard management therapy. The author should mention what further benefit can be achieved from this study.

Response: I completely agree with your point. Despite the WHO recommendation for the application of emollient oil in preterm and low birth weight babies, it is often overlooked or not incorporated into treatment or management guidelines in different countries, including Ethiopia as an example.

The findings of our study, which provide insights into the best type of emollient oil, appropriate dosage, frequency, duration, and potential adverse events associated with its application, highlight the importance of local policy makers taking action. In the Ethiopian context, for instance, it is vital for the Ministry of Health and other relevant stakeholders to support the implementation of emollient oil application as a routine care strategy for preterm infants.

By incorporating this evidence-based intervention into local guidelines and policies, healthcare providers can enhance the care provided to preterm infants, potentially leading to improved outcomes and reduced complications. It is crucial to advocate for the inclusion of emollient oil application as a standard practice, ensuring that preterm infants receive the benefits of this intervention as part of their routine care.

4. The study included 17 articles published between 2000 to July 2023. But I wonder if they missed more important RCTs on this topic that were published in this time frame. For example, Kumar V et al, 2022 (doi: 10.1093/ajcn/nqab430.); Sankaranarayanan K, et al., 2005, PMID: 16208048; Kumar J et al, 2012 (DOI: 10.1007/s12098-012-0869-7). Authors should revise their search and include all missed articles.

Response: Apologies for the confusion caused by the citation issue. We would like to clarify that the study conducted by K. Sankaranarayanan was indeed included in our final analysis. The problem arose with the citation, as the corresponding author's name, Mankir, was mistakenly listed as the first author. However, we have rectified this error in the main document by providing the accurate and corrected citation.

Regarding the study by Kumar V et al., 2022, it was appropriately excluded during the full-text review because it involved term babies and compared the efficacy of sunflower oil versus mustard oil, which did not align with our study's focus on routine care or no oil application in preterm infants. Therefore, it did not meet our strict eligibility criteria.

However, we want to acknowledge that we made an oversight in initially excluding the study conducted by Kumar J et al., 2012 from our final analysis. We take full responsibility for this error, and we have now incorporated it as the eighteenth study in our analysis. By reanalyzing the entire analysis section, we ensure that our findings are robust, comprehensive, and reflect the complete body of relevant literature.

We genuinely appreciate your critical review of our work, as it has helped us identify and address these issues.

5. The authors did the pooled analysis on the mean weight gain after emollient therapy, and it showed significant results. However, their studies were highly heterogeneous. They differ in terms of emollient type, duration and dose. Thus, it may reduce the strength of the study results. Although the authors did subgroup analysis on the pooled analysis, the sample size is too low to strengthen the results. For example, the coconut oil group had two studies, 5ml/kg/day oil group had two studies.

Response: Thank you for highlighting the issue of heterogeneity in the studies included in our analysis. We fully acknowledge this concern, and to address it, we conducted sensitivity analyses, subgroup analyses, and meta-regression. These additional analyses aimed to explore potential sources of heterogeneity and provide a more nuanced understanding of the findings. Furthermore, in light of the heterogeneity observed, we have included recommendations for future research in our study. We emphasize the need for well-designed randomized controlled trials that adhere to standardized protocols.

6. The authors did a quality evaluation of the studies by using the Cochrane Risk of Bias Tool for Randomized Controlled Trials (Higgins et al., 2011), which is an old tool. The ‘RoB 2’ has been revised in August 2019. So, the author should use the updated tool for a better acceptability of their methodology.

Response: We greatly appreciate your valuable feedback. Taking into consideration your insightful suggestions, we have already taken the necessary steps to update our risk of bias assessment tool to the 'RoB 2' version of the Cochrane Risk of Bias Tool for RCTs.

7. As the authors mentioned, the weight gain is multifactorial. Yes, it depends on the combination of the type of emollient oil, its dose and duration. Pooled analysis on homogenous methodological studies can further refute the findings of this study. Thus, research priority has been set to assess the efficacy of the best emollient oil across all available oils for a specific dose and duration. The authors should describe this point, describe the weakness of the heterogeneous study analysis, and come up with some recommendations for future RCTs.

Response: We sincerely value your valuable feedback and suggestions. Based on your insightful input, we have made significant revisions to our manuscript, specifically heterogeneity of studies and the need for further research.

Responses for Reviewer two

1. The authors of ‘Effect of Topical Emollient Oil Application on Weight Gain of Preterm Newborns: A Systematic Review and Meta-Analysis’ have worked on a relevant topic and the review is well performed but my major concern is that this would need thorough language correction and editing.

Response: We would like to express our gratitude for recognizing the relevance and quality of our work. We highly appreciate your positive feedback.

We acknowledge your valid concern regarding the need for thorough language correction and editing in our manuscript. We want to assure you that we have made diligent efforts to address this issue. We have taken your feedback into careful consideration and have allocated sufficient time and resources to perform a comprehensive language correction and editing process.

2. Other comments

The background does not cite all the previous systematic reviews that have been done for this intervention looking at weight and also other outcomes.

Rationale for this systematic review needs to be strengthened

Methods should start with objectives

The authors should mention that how many authors were contacted for full texts and missing data and how many responded

Was ROB 1 used or ROB 2 used for quality assessment.

Methods should define the planned sub-groups

What was the time or age at which weight was measured?

For conclusion, the authors need to make the clinical relevance of about 50g gain in weight and also mention that over what time this change occurred.

Response: We greatly appreciate your valuable feedback and suggestions. We have carefully reviewed and incorporated all of your insightful comments into our manuscript.

Your input has played a crucial role in improving the overall quality and clarity of our work. We have thoroughly revised the manuscript, addressing each of the concerns and suggestions you raised.

---

## [Decision Letter · Decision Letter 1]

11 Apr 2024

PONE-D-24-05779R1EFFECT OF TOPICAL EMOLLIENT OIL APPLICATION ON WEIGHT GAIN OF PRETERM NEWBORNS: A SYSTEMATIC REVIEW AND META-ANALYSISPLOS ONE

Dear Dr. Getaneh,

Thank you for submitting your manuscript to PLOS ONE. After careful consideration, we feel that it has merit but does not fully meet PLOS ONE’s publication criteria as it currently stands. Therefore, we invite you to submit a revised version of the manuscript that addresses the points raised during the review process.

I would like to thank the authors for revising the draft based on reviewers feedback. The draft reads significantly better, however, I think that there are still three major points for authors consideration before the manuscript can be considered for publication. I would suggest that the authors address the three major concerns under the heading of 'Overall comments' and also consider some minor revisions below:

OVERALL: 

1. I would like the authors to address one of the comments from the reviewers on the clinical significance of the difference in mean weight of 52 gms from a policy perspective. The reviewer commented that "I would like the authors to still make the clinical relevance of about 50g gain in weight and also mention that over what time this change occurred. This is important before advocating for its recommendation." Authors can disucss this finding in the 'Discussion' section and frame the conclusion accordingly.

2. In the abstract and the methodology section, the authors have referred to the outcome of interest as 'weight gain' rather than 'weight'. I suggest that the authors refer to the outcome of interest as 'weight' since the outcome of interest is 'weight' and not 'weight gain'. 'weight gain' was a finding of the review.

3. The PRISMA checklist requires the authors to report assessments of certainty (or confidence) in the body of evidence for each outcome assessed. However, the authors have not reported GRADE certainity of the outcomes. I would suggest that the authors assess and report the certainity of evidence for the outcomes of interest.

We look forward to receiving your revised manuscript.

Kind regards,

Rehana Abdus Salam

Academic Editor

PLOS ONE

Additional Editor Comments:

I would like to thank the authors for revising the draft based on reviewers feedback. The draft reads significantly better, however, I think that there are still three major points for authors consideration before the manuscript can be considered for publication. I would suggest that the authors address the three major concerns under the heading of 'Overall comments' and also consider some minor revisions below:

OVERALL:

1. I would like the authors to address one of the comments from the reviewers on the clinical significance of the difference in mean weight of 52 gms from a policy perspective. The reviewer commented that "I would like the authors to still make the clinical relevance of about 50g gain in weight and also mention that over what time this change occurred. This is important before advocating for its recommendation." Authors can disucss this finding in the 'Discussion' section and frame the conclusion accordingly.

2. In the abstract and the methodology section, the authors have referred to the outcome of interest as 'weight gain' rather than 'weight'. I suggest that the authors refer to the outcome of interest as 'weight' since the outcome of interest is 'weight' and not 'weight gain'. 'weight gain' was a finding of the review.

3. The PRISMA checklist requires the authors to report assessments of certainty (or confidence) in the body of evidence for each outcome assessed. However, the authors have not reported GRADE certainity of the outcomes. I would suggest that the authors assess and report the certainity of evidence for the outcomes of interest.

ABSTRACT:

1. The last sentence of the background and first sentence of the methods section are redundantly stating the objective of the review. I would suggest that the authors state the objective only once (either in background or methods) in the abstract.

2. I would suggest rephrasing 'pooled mean weight gain difference' to 'pooled difference in mean weight gain'.

INTRODUCTION:

1. I would suggest authors use standard format of referencing when referring to the previous review, hence either use the last name of the first author (Salam R.A etal) or remove the review authors' name from the sentence to state that "A systematic review conducted in 2013 found that emollient therapy......".

2. I would suggest that the authors move the refereence to the previous systematic review by Salam et al. at the end of the third paragraph to bring all the evidence together for the reader to get an overall background.

METHODS:

1. Correct: 'Publication status: published articles or grey literature'

2. Correct:'the outcome of interest: all research articles that investigated the effect of emollient oil on preterm neonates' weights were included.'

3. Correct: 'Outcome - weight'

4. Correct: 'Additionally, cross-reference search was carried out to include other related studies from ......'.

5. Please add a reference for the Rayan online software.

6. I would suggest that the authors add references to the six articles excluded from the final analysis because the unavailability of the full texts.

7. Primary outcome: 'Estimate the effect of emollient application on weight in preterm newborns (mean difference in grams).'

RESULTS:

1. Correct: 'The primary focus of these studies was to examine the effect of emollient oil therapy on weight in preterm neonates'

2. Characteristics of included studies: I would suggest that the authors specify the range of the mean gestational age as well as the baseline mean weight of the preterm babies from the included studies. These could potentially add context to the heterogeinity in the results.

3. I would suggest that the authors incorporate evidence certainity in the results section.

DISCUSSION:

1. I would suggest authors add some discussion around certainity of evidence for the outcomes reproted.

Reviewers' comments:

Reviewer's Responses to Questions

**Comments to the Author**

1. If the authors have adequately addressed your comments raised in a previous round of review and you feel that this manuscript is now acceptable for publication, you may indicate that here to bypass the “Comments to the Author” section, enter your conflict of interest statement in the “Confidential to Editor” section, and submit your "Accept" recommendation.

Reviewer #1: All comments have been addressed

Reviewer #2: All comments have been addressed

2. Is the manuscript technically sound, and do the data support the conclusions?

Reviewer #1: Yes

Reviewer #2: Yes

3. Has the statistical analysis been performed appropriately and rigorously? 

Reviewer #1: I Don't Know

Reviewer #2: Yes

4. Have the authors made all data underlying the findings in their manuscript fully available?

Reviewer #1: Yes

Reviewer #2: Yes

5. Is the manuscript presented in an intelligible fashion and written in standard English?

Reviewer #1: Yes

Reviewer #2: Yes

6. Review Comments to the Author

Reviewer #1: (No Response)

Reviewer #2: The authors have significantly addressed the comments and improved the manuscript.

I would like the authors to still make the clinical relevance of about 50g gain in weight and also mention that over what time this change occurred. This is important before advocating for its recommendation.

7. PLOS authors have the option to publish the peer review history of their article (what does this mean?). If published, this will include your full peer review and any attached files.

Reviewer #1: No

Reviewer #2: No

---

## [Author Response · Author response to Decision Letter 1]

12 Apr 2024

Point By Point response

We would like to express our deepest appreciation to the editors for their diligent management of the peer review process. Their careful oversight and guidance have been instrumental in ensuring the integrity and quality of our work. We would also like to extend our heartfelt gratitude to the esteemed reviewers who invested their valuable time and expertise in reviewing our papers and providing insightful suggestions. We are grateful for their contributions, and we have thoroughly addressed all their concerns and incorporated their feedback to enhance the overall quality of our research.

Reviewer major concerns:

1. I would like the authors to address one of the comments from the reviewers on the clinical significance of the difference in mean weight of 52 grams from a policy perspective. The reviewer commented that "I would like the authors to still make the clinical relevance of about 50g gain in weight and also mention that over what time this change occurred. This is important before advocating for its recommendation." Authors can discuss this finding in the 'Discussion' section and frame the conclusion accordingly.

Response: Thank you for your feedback. In the revised Discussion section of our manuscript, we have devoted explicit attention to addressing the clinical significance of weight gain and the associated policy implications. Additionally, within the "Characteristics of Included Studies" section, we have incorporated regarding the duration of time during which the observed weight gain occurred.

2. In the abstract and the methodology section, the authors have referred to the outcome of interest as 'weight gain' rather than 'weight'. I suggest that the authors refer to the outcome of interest as 'weight' since the outcome of interest is 'weight' and not 'weight gain'. 'weight gain' was a finding of the review.

Response: We appreciate your suggestion, and we have made the necessary revisions to ensure consistent terminology throughout the entire manuscript.

3. The PRISMA checklist requires the authors to report assessments of certainty (or confidence) in the body of evidence for each outcome assessed. However, the authors have not reported GRADE certainity of the outcomes. I would suggest that the authors assess and report the certainity of evidence for the outcomes of interest.

Response: We have taken your suggestion into consideration and performed a GRADE assessment to evaluate the certainty of evidence for the outcomes of interest. The results of this assessment have been incorporated into the conclusion and discussion section of the revised manuscript.

minor concerns:

Response: We have made thorough efforts to address all the minor concerns raised by the esteemed reviewers. Their feedback was carefully reviewed, and we made the necessary adjustments to the manuscript to ensure that each concern was effectively addressed.

4. Characteristics of included studies: I would suggest that the authors specify the range of the mean gestational age as well as the baseline mean weight of the preterm babies from the included studies. These could potentially add context to the heterogeneity in the results.

Response: Thank you for your concerns. In order to provide clarification on these points, we acknowledge that some of the included studies reported gestational age ranges (e.g., 32-36 weeks) or used terminology such as "<37 weeks," which can limit our ability to calculate precise mean values. Additionally, we understand that certain studies may have only reported the mean weight difference without providing specific baseline weight data for both intervention and comparison groups, making it difficult to include them in our subgroup analysis.

Given these limitations in the available data, we have provided transparency regarding these issues in the "Characteristics of Included Studies" section of the manuscript. We highlight the challenges in computing precise mean gestational age and baseline weight due to the variability in reporting across the included studies.

---

## [Editor Report · Decision Letter 2]

17 Apr 2024

EFFECT OF TOPICAL EMOLLIENT OIL APPLICATION ON WEIGHT OF PRETERM NEWBORNS: A SYSTEMATIC REVIEW AND META-ANALYSIS

PONE-D-24-05779R2

Dear Dr. Getaneh,

We’re pleased to inform you that your manuscript has been judged scientifically suitable for publication and will be formally accepted for publication once it meets all outstanding technical requirements.

Kind regards,

Rehana Abdus Salam

Academic Editor

PLOS ONE
---

## [Editor Report · Acceptance letter]

26 Apr 2024

PONE-D-24-05779R2 

PLOS ONE

Dear Dr. Getaneh, 

I'm pleased to inform you that your manuscript has been deemed suitable for publication in PLOS ONE. Congratulations! Your manuscript is now being handed over to our production team.

Kind regards, 

on behalf of

Dr. Rehana Abdus Salam 

Academic Editor

PLOS ONE